# FRADE: Forgery-aware Audio-distilled Multimodal Learning for Deepfake Detection

Fan Nie
School of Computer Science and
Engineering, Sun Yat-Sen University
Pengcheng Laboratory
Shenzhen, China
nief6@mail2.sysu.edu.cn

Jiangqun Ni*
School of Cyber Science and
Technology, Sun Yat-Sen University
Pengcheng Laboratory
Shenzhen, China
issjqni@mail.sysu.edu.cn

Jian Zhang
School of Computer Science and
Engineering, Sun Yat-Sen University
Guangzhou, China
zhangj299@mail2.sysu.edu.cn

Bin Zhang*
Pengcheng Laboratory
Shenzhen, China
bin.zhang@pcl.ac.cn

Weizhe Zhang
School of Cyberspace Science, Harbin
Institute of Technology
Pengcheng Laboratory
Shenzhen, China
wzzhang@hit.edu.cn

## Abstract

Nowadays, the abuse of AI-generated content (AIGC), especially the facial images known as deepfake, on social networks has raised severe security concerns, which might involve the manipulations of both visual and audio signals. For multimodal deepfake detection, previous methods usually exploit forgery-relevant knowledge to fully finetune Vision transformers (ViTs) and perform cross-modal interaction to expose the audio-visual inconsistencies. However, these approaches may undermine the prior knowledge of pretrained ViTs and ignore the domain gap between different modalities, resulting in unsatisfactory performance. To tackle these challenges, in this paper, we propose a new framework, i.e., **F**orgery-awa**r**e **A**udio-**d**istill**e**d Multimodal Learning (FRADE), for deepfake detection. In FRADE, the parameters of pretrained ViT are frozen to preserve its prior knowledge, while two well-devised learnable components, i.e., the Adaptive Forgery-aware Injection (AFI) and Audio-distilled Cross-modal Interaction (ACI), are leveraged to adapt forgery relevant knowledge. Specifically, AFI captures high-frequency discriminative features on both audio and visual signals and injects them into ViT via the self-attention layer. Meanwhile, ACI employs a set of latent tokens to distill audio information, which could bridge the domain gap between audio and visual modalities. The ACI is then used to well learn the inherent audio-visual relationships by cross-modal interaction. Extensive experiments demonstrate that the proposed framework could outperform other state-of-the-art multimodal deepfake detection methods under various circumstances.

*Co-corresponding author

## CCS Concepts

- **Security and privacy → Social aspects of security and privacy**.

## Keywords

Deepfake Detection, Multimodal Learning, Vision Transformer

**ACM Reference Format:**
Fan Nie, Jiangqun Ni, Jian Zhang, Bin Zhang, and Weizhe Zhang. 2024. FRADE: Forgery-aware Audio-distilled Multimodal Learning for Deepfake Detection. In *Proceedings of the 32nd ACM International Conference on Multimedia (MM '24), October 28–November 1, 2024, Melbourne, VIC, Australia.* ACM, New York, NY, USA, 10 pages. https://doi.org/10.1145/3664647.3681672

## 1 Introduction

The rapid advancement of deep generative models has made the audio-visual effect of AIGC techniques [16, 38, 50, 54] increasingly realistic. However, malicious attackers exploit a specific form of these techniques known as Deepfake to generate and deliver spurious information, thereby posing severe threats to cyber security and personal reputation. With the purpose of eliminating the abuse of deepfakes, it is urgent to develop effective deepfake detection methods.

Previous arts [1, 7, 10, 12, 55] have made much effort on unimodal deepfake detection, e.g., frame- and video-based detection methods, attempting to explore forgery artifacts from a specific modality. However, multimedia content highly depends on audio and visual media forms to convert vivid information, indicating malicious attackers could subtly align multiple forged media to deliver intended information. Ignoring the complementarity of audio-visual inconsistencies within multimedia content, such unimodal detectors suffer from significant performance degradation when applied to multimodal deepfake detection.

Accordingly, some recent studies attempt to exploit audio-visual inconsistencies as cross-modal artifacts to detect multi-modal deepfakes. To effectively mine cross-modal artifacts, [2, 32, 37, 47, 48, 56] extract audio and visual intra-modal features by individual feature backbones and combine them in high-level feature space as global

multimodal features. Nevertheless, intra-modal forgery artifacts mostly cluster in various locations, which is termed the locality of artifacts [1, 7, 10, 11]. Such locality suggests that local audio-visual interaction should be involved to delve into more essential local cross-modal artifacts. Moreover, [17, 47, 52] leverage the pretrained Vision Transformer (ViT) [6, 26] to explore more discriminative audio-visual inconsistencies locally. Despite their superior representation capability, these ViT-based detectors still have the following drawbacks: (1) These detectors are initialized with ViT parameters pretrained on large-scale datasets, e.g., ImageNet [3] and JFT[44], before being fully finetuned on deepfake datasets to explore forgery artifacts. However, finetuned with limited deepfake samples, the ViT-based detectors would forget their pretrained prior knowledge and overfit specific forgery artifacts, which hinders ViTs' generalization capacity to mine unknown forgery artifacts effectively. (2) Although cross-modal interaction between audio-visual modalities is important in capturing cross-modal artifacts, existing methods ignore the domain gap between audio-visual modalities. Therefore, direct and simple audio-visual interaction could impose undesirable modality-specific information into cross-modal forgery features, leading to a suboptimal detection performance.

Motivated by the above analysis, we propose a Forgery-aware Audio-distilled Multimodal Learning (FRADE) framework for detecting audio-visual deepfakes. Specifically, as illustrated in Figure 1, to preserve prior knowledge of pretrained ViTs on large-scale datasets, we maintain the pretrained parameters as frozen at the training phase. For introducing forgery-relevant knowledge into ViTs, we design targeted modules: (1) **Adaptive Forgery-aware Injection.** We explore the high-frequency prior of audio and visual artifacts, and based on the prior, incorporate a few learnable parameters to facilitate intra-modal forgery representation, which is further injected into pretrained ViTs via adaptive transformation into the offsets of corresponding query, key, and value embeddings. These forgery-aware offsets further incorporate pretrained embeddings in the self-attention layer and achieve forgery-relevant knowledge injection. (2) **Audio-Distilled Cross-modal Interaction.** Since the domain gap between audio-visual modalities caused by modality-specific information prevents cross-modal interaction from modeling inherent audio-visual relationships, we propose an audio-distilled interaction to filter modality-specific information before performing cross-modal interaction. Specifically, a set of learnable latent tokens serves as audio-to-visual intermediate tokens to first distill modality-agnostic information from audio features and generate intermediate audio-distilled features, which contain less modality-specific information and are more desirable to be further utilized in cross-modal interaction with visual features. These two modules are equipped with every transformer block in ViT and facilitate the model in learning more discriminative audio-visual representation for deepfake detection.

Overall, our contributions are summarized as follows:

- A novel audio-visual deepfake detection framework, i.e., FRADE, is proposed, which preserves the prior knowledge of ViTs and exploits learnable parameters to effectively capture discriminative intra-modal and cross-modal forgery artifacts and achieve general deepfake detection.

- The Adaptive Forgery-aware Injection (AFI) is devised to inject forgery-relevant knowledge by learning intra-modal forgery features for both audio and visual modalities, while the Audio-Distilled Cross-modal Interaction (ACI) is designed to bridge the domain gap of audio-visual modalities and thus capture crucial audio-visual inconsistencies via cross-modal interaction.

- Experimental results demonstrate the effectiveness and scalability of the proposed modules and the superior detection performance under various evaluation settings.

## 2 Related Work

### 2.1 Unimodal Deepfake Detection

Most existing unimodal methods exploit visual artifacts caused by imperfect forgery techniques to perform effective forensics. For example, [41] leverages some off-the-shelf CNN models to learn forgery clues. Moreover, researchers explore more substantial forgery clues, such as fine-grained local artifacts [1, 7, 10], high-frequency abnormalities [18], and facial blending inconsistencies [27, 42], to achieve better generalization and robustness performance. With the aid of temporal inconsistencies rooted by deepfake generation in a common frame-by-frame manner, [10, 11] are proposed to mine spatiotemporal clues. For detecting audio deepfake, most methods [4, 21] utilize CNN architectures to explore audio spectrum abnormalities. [20, 22] analyze raw audio signals with temporal modeling networks and capture temporal artifacts. Overall, the methods above still treat audio and visual deepfake detection as independent research fields and ignore their correlation. It is crucial to capture the traces of audio-visual discrepancies for detecting advanced audio-visual aligned forgery multimedia, in which both audio and visual deepfakes might appear.

### 2.2 Audio-Visual Deepfake Detection

Initially, researchers capture audio and visual forgery artifacts separately and determine whether the audio-visual has been forged based on the final logits. However, without adequate audio-visual interaction, such methods [2, 56] hardly exhibit multimodal potentials and fail to detect forged videos containing well-constructed audio and corresponding visual contents. To alleviate this problem, later works [31, 32, 47, 48] make a great effort to design multimodal interaction and audio-visual joint learning to mine meaningful discrepancies between the two modalities. Moreover, [8] pretrains the audio-visual model on a real large-scale audio-visual dataset in a self-supervised manner to better capture the cross-modal correlation of real audio-visual samples and achieve promising generalization performance. However, the gap between modalities inevitably introduces interaction-irrelevant modality-specific information into cross-modal interaction, damaging the effect of cross-modal forgery clues. Here, we propose the ACI to eliminate the negative effect of interaction-irrelevant audio features with less interfered modality-specific information then are utilized to perform cross-modal interaction with the visual features.

## 2.3 Audio-Visual Representation Learning

Recently, researchers have transformed the audio signal into a 2D spectrogram (audio image) and proved that pretrained ViT architectures [45, 47] could simultaneously process audio and visual images to extract task-specific features. To fully exploit prior knowledge of pretrained parameters, various adapter modules [15, 28, 39] are designed. Inspired by the above, [34] introduces forgery-related adapters to detect visual forgery artifacts and achieves promising performance. However, such design is customized for unimodal settings, and audio-visual forgery adapters are not fully explored. Here, we propose two types of adapters to facilitate ViTs in mining the audio-visual forgery artifacts, i.e., AFI and ACI. The former is designed to introduce intra-modal forgery knowledge with learnable parameters, while the latter attempts to bridge the domain gap between audio-visual modalities via a few learnable intermediate tokens and then captures more discriminative artifacts by performing cross-modal interaction.

## 3 Method

This section presents our proposed Forgery-aware Audio-Distilled Multimodal Learning (FRADE) framework that adapts frozen pretrained ViTs to audio-visual deepfake detection by introducing forgery-relevant knowledge with a few additional trainable parameters. Concretely, as illustrated in Figure 1, two proposed adapter modules, i.e., Adaptive Forgery-aware Injection (AFI) module and Audio-Distilled Cross-modal Interaction (ACI) module, are employed in every transformer block of a frozen ViT. Note that the frozen parameters of ViT are shared for both the audio and visual inputs.

### 3.1 Audio-Visual Input Embeddings

*Audio and Visual Inputs.* We consider a video clip $V \in \mathbb{R}^{T \times H_v \times W_v \times 3}$ as the visual input, containing $T$ RGB frames with spatial dimensions $H_v \times W_v$. For audio modality, we sample the corresponding audio signal and transform it into 2D spectrogram $A \in \mathbb{R}^{H_a \times W_a}$.

*Audio and Visual Tokenization.* We decompose each RGB frame of the clip $V$ into $m$ non-overlapping patches with the shape $P \times P$ and further transform patches into $D$-dim visual embedding sequence $x_v \in \mathbb{R}^{N_v \times D}, N_v = m \cdot T$. Similarly, we first duplicate the audio spectrogram and stack it as $\tilde{A} \in \mathbb{R}^{H_a \times W_a \times 3}$, which is further decomposed and projected into the audio embedding sequence $x_a \in \mathbb{R}^{N_a \times D}$.

### 3.2 Adaptive Forgery-aware Injection

Since ViTs pretrained on large-scale visual datasets learn the generalized representation, to preserve such representation capability and adapt forgery-relevant representation learning in audio (or visual) modality, we design the Adaptive Forgery-aware Injection (AFI) module, in which the following factors are taken into account.

*Injection Feasibility.* We expect the proposed AFI to be capable of mining both audio and visual artifacts within an identical design, which facilitates subsequent audio-visual embedding fusion in exploring cross-modal artifacts of the audio-visual. Plenty of works point out the effectiveness of high-frequency artifacts in the visual. For the audio, we would visualize and analyze real and fake audio

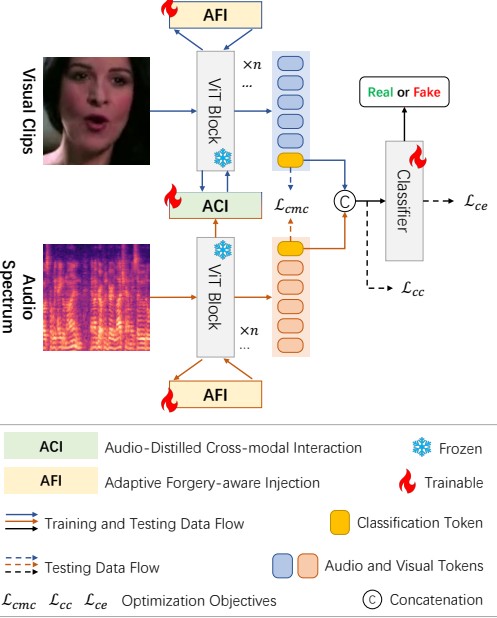

**Figure 1: Overview of the proposed FRADE. Initially, the audio-visual inputs are sliced and embedded into sequences. They are fed into stacked transformer blocks with AFI and ACI modules. Finally, the MLP classifier determines their authenticity.**

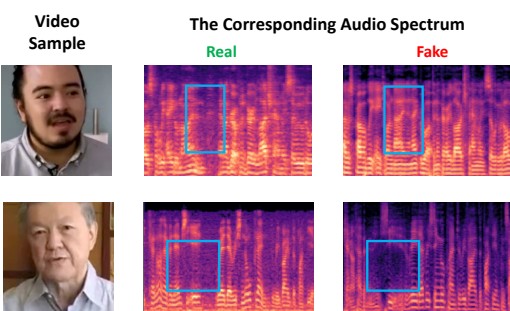

**Figure 2: Illustration of the audio spectrogram from the perspective of the image.**

spectrograms from the perspective of the spatial image, which is regarded as the texture image when fed into the ViT embedding layer. As illustrated in Fig. 2, the texture difference between real and fake spectrograms of the same visual content (see it in blue boxes) is mainly clustered in high-frequency contents, i.e., highly textual regions. Therefore, when designing the AFI module, we expect it to focus on capturing high-frequency abnormalities and achieve intra-modal forgery representation learning within the identical structure.

*Injection Location.* Every transformer block of ViTs has two core components: Multi-Head Self-Attention (MHSA) and Feed-Forward Network (FFN). Both components can be used to integrate extracted

task-related knowledge into the prior [28, 39]. Moreover, compared with FFN, MHSA provides a more complex non-linear latent space, which has a better representation capability and enables the module to extract more subtle but crucial forgery artifacts. Therefore, based on the above analysis, we deploy the proposed AFI module before the MHSA to fully exploit the capability of self-attention and capture more discriminative forgery artifacts.

Guided by the above intuitions, we propose the AFI module, depicted in Figure 3. To facilitate the injection of forgery-relevant knowledge in terms of high-frequency prior, a few learnable parameters are introduced to serve as the offsets of query, key, and value embeddings in MHSA. Specifically, with the pretrained weights of MHSA kept frozen, AFI exploits a predefined high-pass filter to suppress low-frequency components and extracts forgery artifacts, which mainly concentrate on high-frequency components. Then, the extracted features are transformed into the offsets of the query (q), key (k), and value(v) embeddings, which are integrated with the original embeddings of MHSA, achieving the knowledge injection of intra-modal forgery artifacts.

Initially, the input audio (or visual) sequence $x \in \mathbb{R}^{N \times D}$ is reshaped into the spatial form $\hat{x} \in \mathbb{R}^{\hat{H} \times \hat{W} \times D}$ with $(\hat{H}, \hat{W}) = (\frac{H}{P}, \frac{W}{P})$. Here, $(H, W) = (H_a, W_a)$ for the audio input $x_a$. Then, we employ convolution with $3 \times 3$ kernels to learn the relationships among neighboring patches and capture regional and local forgery artifacts. Subsequently, for capturing forgery artifacts existing at the high-frequency domain, we incorporate the Fast Fourier Transformation (FFT) and the inverse (IFFT) with a predefined low-frequency filtering mask $M_f$ to filter low-frequency semantic information and guide the detector to focus on high-frequency abnormal contents, which include more obvious artifacts [29, 36]. Lastly, we project the filtered features into the forgery-aware offsets of $q$, $k$, and $v$ embeddings, which are integrated with the corresponding embeddings for MHSA. The process of the AFI module is formulated as follows:

$$x_f = \text{IFFT}(M_f * \text{FFT}(\text{Conv}_{3 \times 3}(\hat{x}))), \quad (1)$$

$$\Delta q, \Delta k, \Delta v = \text{Conv}_{1 \times 1}(x_f), \text{Conv}_{1 \times 1}(x_f), \text{Conv}_{1 \times 1}(x_f), \quad (2)$$

$$\hat{x}_{out} = \text{Softmax}(\frac{(q + \Delta q) \cdot (k + \Delta k)^\top}{\sqrt{D}})(v + \Delta v), \quad (3)$$

where $\Delta q$, $\Delta k$, and $\Delta v$ represent the forgery-aware offsets. In particular, depending on the characteristics of the component distribution in frequency domain, the shape of $M_f$ is $\hat{H} \times \hat{W}$ and values are initialized as one, where the central area with radius $d_f$ is set as zero to form the high-pass filter. Note that MHSA performs multi-head projection on input sequences $x$ in the generation of query, key, and value embeddings. Therefore, we also project the intermediate features $x_f$ into the corresponding offsets in the multi-head form. By performing AFI on both audio and visual inputs, we obtain their intra-modal embeddings.

### 3.3 Audio-Distilled Cross-modal Interaction

Furthermore, multimodal embedding fusion is performed on the audio and visual sequences to learn the inherent correlations of both modalities jointly. For audio-visual deepfake detection, previous arts commonly realize the negative effect of the domain gap

between audio-visual forgery representations and design more sophisticated fusion strategies, e.g., cross attention [47, 52], weighted aggregation with contrastive learning [32, 48], to bridge this gap and capture more substantial cross-modal artifacts. However, existing methods implicitly enclose domain gaps via complex optimization objectives, which results in the detector overfitting in specific audio-visual inconsistencies and being trapped in the local optimal.

Here, we propose the ACI to explicitly enclose the domain gap via learnable parameters rather than customized optimization objectives and perform cross-modal interaction. Specifically, before interacting with visual embeddings, audio embeddings are distilled with a few intermediate tokens via cross-attention operations to extract interaction-specific audio information while discarding modality-specific information, which effectively encloses the domain gap between modalities. Note that, benefitting from the hierarchical design of general ViTs, the proposed FRADE could progressively fuse visual and distilled audio features via a series of ACI modules and thoroughly model audio-visual relationships, thus capturing more intrinsic artifacts.

As illustrated in Figure 3, in the ACI module, we predefine a few intermediate learnable tokens $x_{inter} \in \mathbb{R}^{N_{inter} \times D}$. $N_{inter}$ is a hyperparameter that controls the number of learnable tokens. Subsequently, given the audio-visual sequence $x_a$ and $x_v$, $x_{inter}$ distills interaction-specific information $\hat{x}_{inter}$ from the original audio sequence via cross-attention interaction. Finally, we perform cross-attention interaction on the distilled audio tokens $\hat{x}_{inter}$ and visual sequence $x_v$ to further explore cross-modal artifacts. Note that we set $N_{inter}$ much smaller than $N_a$, indicating $x_d$ tends to compress audio features and extract interaction-specific audio information. The progress of the ACI module can be formulated as:

$$\hat{x}_{inter} = x_{inter} + \text{CrossAttn}(x_{inter}, x_a)$$
$$= x_{inter} + \text{Softmax}(x_{inter} \cdot x_a^\top)x_a, \quad (4)$$

$$\hat{x}_v = \text{Conv}(x_v + \text{CrossAttn}(x_v, \hat{x}_{inter}))$$
$$= \text{Conv}(x_v + \text{Softmax}(x_v \cdot \hat{x}_{inter}^\top)\hat{x}_{inter}), \quad (5)$$

where $\text{Conv}(\cdot)$ stands for convolution-related operations, including down-sampling convolution, non-linear projection, and up-sampling convolution operations.

### 3.4 Optimization Objectives

In this section, the audio and visual sequences $x_a, x_v$, processed by intended ViTs, are utilized to predict whether audio-visual contents are forged or not. Concretely, following the general settings of ViT, we perform linear projection on the classification tokens $cls_a, cls_v$ of both sequences into the cross-modal global representation $cls_g$. The global representation $cls_g$ is then exploited to compute the prediction probability $\hat{y}$ for audio-visual deepfake detection.

Furthermore, in the real-world scenario, visual (or audio) or both modalities can be manipulated, indicating there are three forgery cases: RealVisual-FakeAudio (RVFA), FakeVisual-RealAudio (FVRA), and FakeVisual-FakeAudio (FVFA). When detecting RVFA and FVRA, the detector would be misled by the pristine features of the unaltered modality and thus more fine-grained optimization is

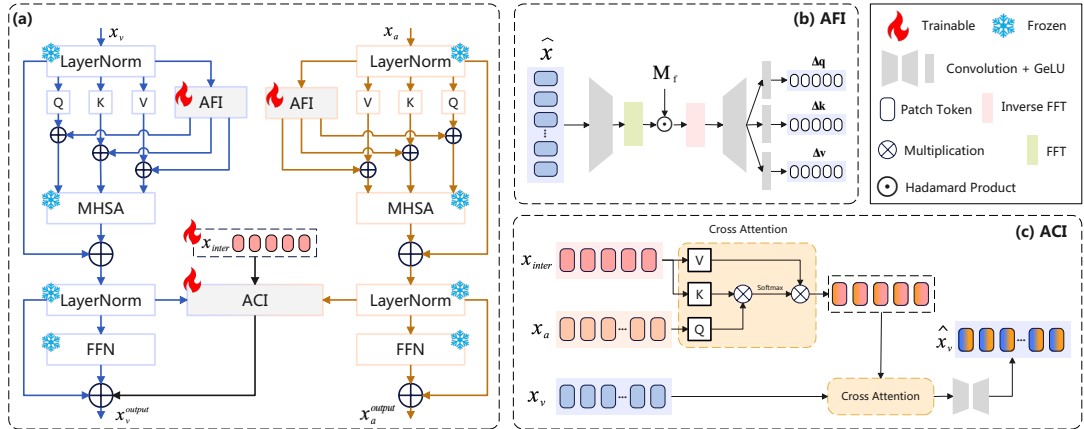

**Figure 3: (a) The audio-visual data flow of one ViT block in the FRADE framework. (b) The Adaptive Forgery-aware Injection (AFI) module. (c) The Audio-distilled Cross-modal Interaction (ACI) module. Note that we exclude the classification tokens before feeding audio-visual sequences into AFI and ACI modules.**

required to tackle this issue. Specifically, for optimizing the single-side forgeries, i.e., RVFA and FVRA, the feature distance between the two modalities is expected to be larger than the real audio-visual pairs (RealVisual-RealAudio, RVRA). On the other hand, to better identify the RVRA samples from the others, we expect RVRA samples to be clustered while being kept a certain distance from the fake samples. Therefore, we design the following loss functions over $N$ training samples to achieve these goals.

**Cross-modal Contrastive Loss** ($\mathcal{L}_{cmc}$) attempts to pull audio and visual representations of RVRA samples closer while pushing them from each other for the RVFA, FVRA, and FVFA. Formally, this process can be denoted as:

$$\mathcal{L}_{cmc} = \frac{1}{N} \sum_{i=1}^{N} [(1 - y_i) \cdot (1 - \text{Sim}(cls_a^i, cls_v^i)) + \qquad (6)$$
$$y_i \cdot max(\text{Sim}(cls_a^i, cls_v^i) - \gamma_1, 0)],$$

where $y_i$ is the ground-truth label of the $i$-th audio-visual sample, $y_i$ is 0 for the RVRA sample, otherwise $y_i$ is 1. Moreover, $\text{Sim}(\cdot)$ denotes the cosine similarity function, and $\gamma_1$ controls the separation degree of fake audio and visual features, set as 0.2.

**Audio-Visual Center Cluster Loss** ($\mathcal{L}_{cc}$) is designed for clustering RVRA samples and facilitates the learning of discriminative audio-visual features. Here, we define a learnable RVRA feature center $f_c \in \mathbb{R}^D$ and perform cluster optimization on cross-modal representation $cls_g$ as follow:

$$\mathcal{L}_{cc} = \frac{1}{2|\Omega_r|} \sum_{i \in \Omega_r} ||cls_g^i - f_c||_2^2 - min(\frac{1}{2|\Omega_f|} \sum_{i \in \Omega_f} ||cls_g^i - f_c||_2^2, \gamma_2),$$
$$(7)$$

where $\Omega_r$ and $\Omega_f$ represent the index sets of real and fake samples, respectively. $\gamma_2$ controls the minimal distance between the RVRA center and fake samples and is set as 0.25.

**Cross-Entropy Loss** ($\mathcal{L}_{ce}$) corresponds to our detection goal and ensures the learning of forgery-relevant features on both modalities.

$$\mathcal{L}_{ce} = -\frac{1}{N} \sum_{i=1}^{N} [y_i \log(\hat{y}_i) + (1 - y_i)\log(1 - \hat{y}_i)]. \qquad (8)$$

In summary, our FRADE framework is trained with the weighted aggregation of Audio-Visual Cross-modal Contrastive Loss ($\mathcal{L}_{cmc}$), Audio-Visual Center Cluster loss ($\mathcal{L}_{cc}$), and Cross-Entropy Loss ($\mathcal{L}_{ce}$) which can be expressed as :

$$\mathcal{L}_{all} = \mathcal{L}_{ce} + \alpha_1 \mathcal{L}_{cmc} + \alpha_2 \mathcal{L}_{cc}, \qquad (9)$$

where $\alpha_1$ and $\alpha_2$ are the hyperparameters to balance the effects of individual loss functions.

## 4 Experiments

### 4.1 Datasets

Following recent audio-visual forensics arts [8, 32, 47, 48], we leverage several public datasets to comprehensively evaluate our method, which compacts a diverse range of scenarios, manipulation techniques, and real-world perturbations. (1) **FakeAVCeleb** [23] contains 500 real videos and 19,500 deepfake videos, where real videos are collected from different individuals of various ethnic groups. FaceSwap [35] and FSGAN [38] are leveraged to generate face-swapped videos, and fake audios are generated by SV2TTS [19] and Wav2Lip [40]. (2) **KoDF** [25] includes 62,166 unique 90-second-long real video clips collected on various conditions and 175,776 deepfake clips of at least 15 seconds, where fake videos are generated by six synthesized methods: FaceSwap, FSGAN, DeepFaceLab [30], FOMM [43], ATFHP [49], and Wav2Lip. The original audio track of each video is unchanged. (3) **DeAVMiT** [47] is a recent multimodal dataset that contains 540 pristine videos and 6480 deepfake videos, in which multiple audio forgery techniques are involved to reduce explicit audio-visual forgery traces. (4) **Deepfake Detection Challenge (DFDC)** [5] is a publicly available dataset with 19,154 real video clips sourced from 3426 actors and more than 100,000

deepfake clips forged by various techniques. To better simulate real-world scenarios, the forged videos are distorted with diverse perturbations such as color saturation, blurring, compression, etc.

## 4.2 Implementation Details

We sample 16 frames ($T = 16$) with 4 intervals for each video as an input clip and extract corresponding faces using MTCNN [51], which are aligned to the size of $224 \times 224$. Meanwhile, similar to [8], we resample the audio signal to $16kHz$ and then transform it into the log-mel spectrogram by short-time Fourier Transform with 80 mel filter banks, a hop length of 160, and a window size of 320. Note that we employ the FRADE framework into the original ViT-base backbone [6] to perform the following comparative evaluations. Furthermore, we utilize the AdamW optimizer with the learning rate 2e-6. For hyperparameter settings, the weight parameters $\alpha_1$, $\alpha_2$ in Eq. (9) are set as 0.3 and 0.4, respectively, while we conduct ablation experiments to determine the optimal $d_f$ and $N_{\text{inter}}$ values.

## 4.3 Evaluation Metrics

Following the previous arts [8, 32, 47, 48], we report the Accuracy (ACC) and Area Under Curve (AUC) as evaluation metrics. Moreover, we sample sequentially all audio-visual pairs from each video during evaluation and average them as recorded performance. Unless otherwise specified in the following comparisons, **red** indicates the best performance on a specific dataset, while **blue** means the second-best performance.

## 4.4 Comparative Experiments

In this section, we utilize four public datasets, including FakeAVCeleb, DeAVMiT, KoDF, and DFDC, to conduct comparative experiments with recent state-of-the-art methods.

*4.4.1 Intra-dataset Evaluation.* Here, we conduct the intra-dataset evaluation within several datasets, i.e., all detectors are trained and tested on the identical dataset. Table 1 demonstrates the comparative results regarding performance metrics ACC and AUC, which reveal the detectors' capability of capturing audio-visual forgery artifacts. Specifically, it is observed as follows: (1) Overall, compared with visual detectors, audio-visual detectors consider more comprehensive audio-visual forgery information and achieve better performance. (2) Moreover, when evaluated on FakeAVCeleb, which contains more diverse audio-visual forgery combinations, most existing detectors obtain relatively inferior performance. We attribute the inferiority to two factors: they ignore the locality of artifacts and leverage high-level features to mine coarse cross-modal artifacts via audio-visual interaction. The domain gap between modalities undermines the effectiveness of audio-visual interaction. In contrast, the proposed method utilizes the stacked ACI modules to progressively bridge the domain gap and fuse audio-visual features and incorporates the AFI module to further mine more instinctive cross-modal artifacts. Therefore, it achieves superior performance across various datasets.

*4.4.2 Cross-dataset Evaluation.* To further evaluate the generalizability of the proposed method, we conduct the cross-dataset evaluation on the detection of unseen datasets following the settings in [32, 47, 48]. Specifically, The comparative results are demonstrated

in Table 2 and Table 3. Note that we borrow the results of counterparts from MCL[32] and compare with them in Table 3, where compared methods are trained on DF-TIMIT [24], besides listed datasets. It is observed that most methods show an obvious performance improvement when trained on more audio-visual pairs, i.e., the evaluation setting of Table 3, and our method achieves superior generalization performance under most of the evaluation settings, about 2.5 % averaged improvement. Such improvement can be attributed to two factors. (1) Frozen parameters of pretrained blocks preserve the generalized feature representation capability, which further constrains the learnable AFI and ACI modules to extract more generalized artifacts. (2) The proposed method exploits audio-visual high-frequency forgery artifacts, which are more feasible to distinguish the forged samples, thereby improving the detector's generalizability.

*4.4.3 Cross-forgery Evaluation.* Similar to the previous art [8], we conduct the experiment on the FakeAVCeleb dataset to evaluate the generalizability of our method against unseen forgery cases. In the cross-forgery scenario, the detectors are required to detect artifacts of unseen forgery cases. To simulate this scenario, we divide FakeAVCeleb samples into four cases: RVRA, RVFA, FVRA, and FVFA, and hold out the evaluated forgery case while training the models on the remaining cases. We show the results in Table 4, and it could be observed as follows. (1) Compared with unimodal detectors, audio-visual joint modeling enables the multimodal detectors to detect RVFA samples and achieve acceptable performance. Meanwhile, naive and simple cross-modal interaction of existing methods leads to performance degradation when detecting visual-related artifacts, i.e., FVRA and FVFA. (2) The cross-modal inconsistencies differ from various forgery cases, i.e., RVFA, FVRA, and FVFA, and require more adequate and fine-grained modeling by cross-modal interaction. Compared with existing audio-visual methods, the design of progressive interaction in FRADE could effectively model audio-visual relationships and better deal with various unseen forgery cases.

## 5 Ablation Studies

## 5.1 Component Contribution

Table 5 illustrates the contribution of each proposed module and loss to overall performance improvement in FRADE. Note that for measuring the contribution of the audio-distilled interaction, we compare the proposed method with its variant, which performs vanilla cross-attention interaction utilized in [47]. It is observed in Table 5 as follows. (1) Both AFI and ACI modules are crucial in modeling audio-visual relationships and further identifying convincing forgery artifacts. (2) Compared with the vanilla interaction, the audio-distilled interaction of the ACI module includes the extra step beside cross-attention interaction, i.e., leveraging intermediate tokens to distill modality-agnostic audio information. This well-designed module effectively encloses the domain gap in audio-visual modalities and thus improves the model's capability of modeling audio-visual relationships. (3) Furthermore, $\mathcal{L}_{cmc}$ and $\mathcal{L}_{cc}$ provide more compact latent space corresponding to robust decision boundaries to the detector. When evaluated on the DeAVMiT dataset, which contains relatively low-quality audio-visual samples, the

**Table 1: Intra-dataset Evaluation. Note that the detector with † represents it belongs to the unimodal method, i.e., visual detector.**

| Method | Venue | FakeAVCeleb | | KoDF | | DFDC | | DeAVMiT | |
|---|---|---|---|---|---|---|---|---|---|
| | | ACC (%) | AUC (%) | ACC (%) | AUC (%) | ACC (%) | AUC (%) | ACC (%) | AUC (%) |
| LipForensics† [14] | CVPR 2021 | 80.1 | 82.4 | 93.16 | 93.72 | 71.3 | 73.5 | 73.1 | 77.2 |
| MultiAttn† [53] | CVPR 2021 | 77.6 | 79.3 | 91.1 | 90.5 | 89.8 | 92.2 | - | - |
| RealForensics† [13] | CVPR 2022 | 90.1 | 92.3 | - | - | 89.6 | 91.5 | 92.8 | 96.2 |
| MDS [2] | MM 2020 | 82.8 | 86.5 | 95.68 | 95.24 | 89.8 | 91.6 | 92.0 | 94.3 |
| Joint-AVD [56] | ICCV 2021 | 82.5 | 83.3 | 92.96 | 93.59 | 90.2 | 91.9 | 89.6 | 93.7 |
| AVFakeNet [17] | ASC 2022 | 78.4 | 83.4 | - | - | 82.8 | 86.2 | 91.8 | 93.7 |
| AVoiD-DF [47] | TIFS 2023 | 83.7 | 89.2 | - | - | 91.4 | 94.8 | 90.1 | 93.9 |
| MCL [32] | TCSVT 2023 | 86.0 | 89.6 | 97.8 | 98.1 | 97.9 | 98.3 | - | - |
| AVAD [8] | CVPR 2023 | 94.2 | 94.5 | 87.6 | 86.9 | 93.2 | 96.7 | 96.3 | 97.7 |
| PVASS-MDD [48] | TCSVT 2023 | 95.7 | 97.3 | - | - | 96.3 | 98.9 | 97.6 | 99.1 |
| FRADE | Ours | 98.6 | 99.8 | 99.1 | 99.8 | 97.2 | 99.0 | 98.8 | 98.6 |

**Table 2: Cross-dataset Evaluation. All detectors are trained on FakeAVCeleb, and tested on the other three datasets, respectively. Results are reported in terms of AUC (%).**

| Method | Venue | KoDF | DeAVMiT | DFDC |
|---|---|---|---|---|
| LipForensics [14] | CVPR 2021 | 86.6 | 52.5 | 53.1 |
| FTCN [55] | ICCV 2021 | 68.1 | - | - |
| MDS [2] | MM 2020 | - | 75.2 | 73.1 |
| Joint-AVD [56] | ICCV 2021 | - | 77.8 | 76.7 |
| AVAD [8] | CVPR 2023 | 86.9 | 83.7 | 81.4 |
| PVASS-MDD [48] | TCSVT 2023 | - | 87.5 | 84.8 |
| AVoiD-DF [47] | TIFS 2023 | - | 83.2 | 80.7 |
| FRADE | Ours | 92.4 | 89.3 | 83.8 |

**Table 3: Cross-dataset Evaluation. All detectors are evaluated on one specific dataset while trained on the remaining datasets. Results are reported in terms of AUC (%).**

| Method | Venue | KoDF | FakeAVCeleb | DFDC |
|---|---|---|---|---|
| LipForensics [14] | CVPR 2021 | 74.7 | 77.8 | 74.3 |
| FTCN [55] | ICCV 2021 | 78.1 | 79.3 | 74.0 |
| MDS [2] | MM 2020 | 79.0 | 79.3 | 72.3 |
| Joint-AVD [56] | ICCV 2021 | 82.5 | 83.5 | 75.7 |
| AVAD [8] | CVPR 2023 | 86.9 | - | 81.4 |
| MCL [32] | TCSVT 2023 | 87.2 | 87.1 | 86.8 |
| FRADE | Ours | 93.5 | 93.1 | 85.4 |

**Table 4: Cross-Forgery Evaluation. All detectors are evaluated on one specific forgery case while trained on the remaining cases. The results are reported in terms of AUC (%).**

| Method | Multimodal | Forgery Case | | |
|---|---|---|---|---|
| | | RVFA | FVRA | FVFA |
| Lipforensics [14] | | - | 97.7 | 88.9 |
| FTCN [55] | | - | 97.4 | 91.6 |
| RealForensics [13] | | - | 93.0 | 98.5 |
| Joint-AVD [56] | ✓ | 73.3 | 97.4 | 85.0 |
| AVBYOL [9] | ✓ | 50.0 | 61.3 | 58.5 |
| AVAD [8] | ✓ | 80.5 | 93.7 | 92.7 |
| FRADE | ✓ | 97.7 | 97.3 | 99.2 |

**Table 5: Component Contribution Evaluation. The FRADE and its variants are trained on the FakeAVCeleb.**

| Variant | | | | FakeAVCeleb | KoDF | DeAVMiT |
|---|---|---|---|---|---|---|
| AFI | ACI | $\mathcal{L}_{cmc}$ | $\mathcal{L}_{cc}$ | | | |
| ✓ | | | | 95.7 | 74.1 | 64.8 |
| ✓ | ✓ | | | 99.7 | 88.7 | 84.4 |
| ✓ | | ✓ | ✓ | 99.2 | 78.5 | 65.3 |
| ✓ | ✓ | ✓ | | 100.0 | 91.5 | 87.7 |
| ✓ | ✓ | | ✓ | 99.5 | 93.2 | 87.3 |
| ✓ | ✓ | ✓ | ✓ | 99.8 | 92.4 | 89.3 |

detector without above optimization objectives, i.e., $\mathcal{L}_{cmc}$ and $\mathcal{L}_{cc}$, suffers from the noisy features and achieve inferior results. Therefore, both optimization objectives are essential for the audio-visual detector to learn a more generalized feature representation.

*The Impact of $d_f$ and $N_{inter}$ values.* In each ViT block of the proposed FRADE, $d_f$ in the AFI module controls the degree of filtering, and the larger $d_f$ value indicates the more intensive filtering effect, which discards wider-range frequency contents. Moreover, $N_{inter}$ in the ACI module determines the number of intermediate tokens for

audio distillation and controls the distillation degree of audio information. The results are depicted in Table 6 and analyzed as follows: (1) When the $d_f$ value grows larger, the corresponding high-pass filter tends to discard more high-frequency information and leads to a severe performance drop on the cross-forgery scenario. (2) Besides, when the $N_{inter}$ grows larger, more modality-specific information is inevitably introduced into distilled tokens, which causes the detector to overfit specific audio-visual forgery clues and thus damages the generalizability of the detector.

*The Impact of the Modality Distillation.* We realize that audio and visual contents differ from the information volume and design

 Fan Nie, Jiangqun Ni, Jian Zhang, Bin Zhang, & Weizhe Zhang

**Table 6: Hyperprameter Evaluation. We utilize various combinations of $N_{\text{inter}}$ and $d_f$ to train the FRADE on FakeAVCeleb.**

| $N_{\text{inter}}$ | $d_f$ | FakeAVCeleb | | | KoDF | DeAVMiT |
|---|---|---|---|---|---|---|
| | | RVFA | FVRA | FVFA | | |
| 2 | 0 | 97.3 | 99.2 | 98.5 | 87.9 | 85.7 |
| 4 | 0 | 98.2 | 98.4 | 99.5 | 90.6 | 86.1 |
| 4 | 7 | 95.2 | 93.8 | 97.6 | 91.8 | 88.4 |
| 4 | 11 | 95.7 | 94.2 | 98.2 | 92.1 | 87.5 |
| 8 | 3 | 98.5 | 97.9 | 99.7 | 89.8 | 90.1 |
| 16 | 3 | 98.4 | 97.6 | 99.9 | 88.2 | 87.3 |
| 4 | 3 | 97.7 | 97.3 | 99.2 | 92.4 | 89.3 |

**Table 7: Modality Distillation Evaluation. We design multiple distillation strategies and evaluate all detectors trained on FakeAVCeleb in terms of AUC (%).**

| Distillation Variant | FakeAVCeleb | KoDF | DeAVMiT |
|---|---|---|---|
| None | 99.7 | 83.4 | 81.8 |
| Visual | 98.9 | 86.3 | 81.4 |
| Audio-Visual | 99.9 | 88.8 | 85.6 |
| Audio | 99.8 | 92.4 | 89.3 |

**Table 8: Scalability Evaluation. We adopt the FRADE framework into different backbones. The AUC (%) scores and backbone parameter volumes 'Param' (million, M) are reported below. And the gray background indicates the default setting.**

| Backbone | Param | FakeAVCeleb | KoDF | DeAVMiT |
|---|---|---|---|---|
| Swin-V2-Base | 88M | 99.5 | 94.1 | 90.5 |
| Swin-V2-Large | 197M | 99.8 | 94.5 | 91.2 |
| ViT- Large | 307M | 99.6 | 92.6 | 87.7 |
| ViT-Base | 86M | 99.8 | 92.4 | 89.3 |

unidirectional interaction, i.e., audio-distilled interaction, in the ACI module. To verify this design, we evaluate multiple distillation variants, including Visual-, Audio-, and bidirectional (Audio-Visual) distilled interactions illustrated in Table 7. Overall, all variants, especially the audio-distilled variant, benefit from the design of modality distillation and obtain the expected performance improvement. We attribute this to the following factors: (1) Modality distillation effectively encloses the domain gap between audio and visual modalities, which is crucial for cross-modal interaction. (2) Moreover, the audio modality is regarded as the time sequence with rich temporal information, which could guide the detector to capture instinct temporal artifacts in the visual modality by cross-modal interaction. In contrast, besides temporal information, the visual forgery modality includes more obvious spatial artifacts, primarily dominant in mining visual forgery clues [46]. That means performing a visual-distilled interaction with audio would disturb the temporal forgery clues of audio, thus causing significant performance degradation.

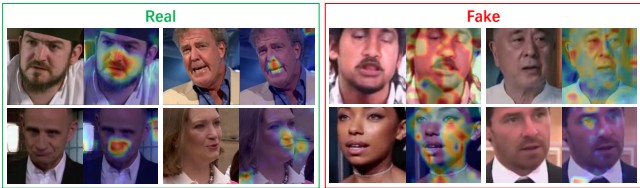

**Figure 4: Visualization of cross-attention maps in the last ACI module. The warmer color indicates a higher response of audio-distilled $\hat{x}_{inter}$ to certain facial regions.**

## 5.2 Scalability

We also emphasize that our FRADE framework is independent of the specific ViT structure and conduct comparative experiments on various ViTs with diverse parameter volumes. As illustrated in Table 8, it is observed that: (1) benefiting from the more sophisticated structure, i.e., the sliced attention window in Swin Transformer [33], our FRADE could achieve better generalization performance, which indicates the FRADE is the backbone-agnostic design and has the potential to adapt to various ViT structures, i.e., more advanced structures bring better forgery representations. (2) Furthermore, more designed modules in larger backbones, i.e., AFI and ACI, with more learnable parameters would result in our FRADE overfitting specific features, limiting its generalizability.

## 5.3 Visualization

To further demonstrate the cross-modal interaction effect of the proposed ACI module in FRADE, we visualize and analyze cross-attention score maps generated in Eq. (5). As illustrated in Figure 4, it is observed that: (1) the learnable intermediate tokens in ACI could effectively distill interaction-specific information from audio and guide the model to focus on certain face regions. (2) Moreover, depending on different data types, i.e., Real and Fake, the tokens would interact with different regions of the visual. For the real audio-visual pairs, the high response concentrates on mouth-nose regions, which are closely related to audio information. It suggests that the learnable tokens could construct real audio-visual relationships via mouth-nose movement. In contrast, fake audio or visual modality would undermine real audio-visual relationships and distract the attention of $\hat{x}_{inter}$ to mouth-nose regions.

## 6 Conclusion

This paper introduces a novel audio-visual deepfake detection framework, i.e., the Forgery-Aware Audio-Distilled Multimodal Learning (FRADE). Specifically, we introduce forgery-relevant knowledge into the ViT backbones via trainable parameters while freezing the pretrained parameters to preserve the general prior knowledge. To effectively capture generalized intra-modal artifacts, we propose an Adaptive Forgery-aware Injection (AFI) module to explore the universal frequency characteristics of audio-visual artifacts. Furthermore, we develop an Audio-Distilled Cross-modal Interaction (ACI) module to bridge the domain gap between audio-visual modalities, enhancing cross-modal forgery interaction. Extensive experiments demonstrate the superiority and scalability of the proposed method.

# Acknowledgments

This work was supported in part by the National Natural Science Foundation of China under Grants U23B2022, U22A2030, the Guangdong Major Project of Basic and Applied Basic Research under Grant 2023B0303000010, and the Major Key Project of PCL under Grant PCL2023A05.

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
