# OpenReview forum: "FRADE: Forgery-aware Audio-distilled Multimodal Learning for Deepfake Detection"
_acmmm.org/ACMMM/2024/Conference — MM2024 Poster_

### Official Review · Reviewer_kdxJ · 2024-05-25

**Rating:** 4
**Confidence:** 3

**Summary:**

The paper introduces a new framework aimed at enhancing Deepfake detection by leveraging both audio and visual modalities. The framework integrates ViT with two main components: Adaptive Forgery-aware Injection (AFI) and Audio-distilled Cross-modal Interaction (ACI). These components are designed to preserve the pretrained ViTs' knowledge while incorporating forgery-relevant information and bridging the domain gap between audio and visual data.

**Strengths:**

1. Experiments on both cross-forgery and cross-dataset scenarios showed improved performance over previous methods, indicating good generalization capability.
2. The ACI module is well designed to discard modality specific information by learning intermediate tokens.
3. Cross-attention maps help in understanding how the model focuses on different regions of the visual data when interacting with audio data.

**Limitations:**

The paper is not well presented and the following issues are unclear.
1. What is the dimension after 3x3 convolution in AFI module? Number of parameters in the designed module might need to be clarified to assess its transferability.
2. In Section 3.2, the authors declared that “compared with FFN, MHSA provides a more complex non-linear latent space, which has a better representation capability and enables the module to extract more subtle but crucial forgery artifacts.”. However, MHSA only has linear operations. And why MHSA has a better representation capability in this task? Please provide more rigorous proof or cite relevant sources.
3. In the Formula (4), it indicates that x_inter is the query. However, in Figure3, x_a is drawn as the query. The authors should avoid such errors.
4. What does x_d mean in Line 443 Page 4?
5. Why are the comparison methods in Tables 2-4 different? Especially the well performing PVASS-MDD did not appear in Tables 3 and 4. Perhaps the authors need to reproduce it for a more comprehensive comparison.

**Suitability:**

3

---

### Official Review · Reviewer_zrJn · 2024-05-25

**Rating:** 4
**Confidence:** 3

**Summary:**

This work aims to solve the previous gap that fine-tuning the pretrained models will destroy their universal abilities. This work presents a new deepfake detection method addressing keeping the universal knowledge in pretrained ViT weights, by proposing 2 modules:

1. Adaptive Forgery-aware Injection (AFI): An adapter mechanism to adapt the frozen ViT for deepfake detection.
2. Audio-Distilled Cross-modal Interaction (ACI): A cross-modal interaction mechanism for analyzing visual/audio inputs.

**Strengths:**

1. This paper clearly explain the method and the motivations behind it.
2. Extensive benchmark with baselines for intra-dataset and cross-dataset are analyzed.

**Limitations:**

1. In Section 4.4.3, when the model is training on partial modality modified samples (RVFA, FVRA), what kind of labels are seen by the model? For example, do you provide “fake” label for RVFA, or modality specified (fake on visual, but real on audio) to the model? If the latter one, I think the 2 modalities are work “nearly-independent”, and the experiments cannot show the effectiveness of multimodal communications. What if only one video-level labels are provided and see if the audio/visual modules can still figure the real and fake for each modality?
2. The majority experiments are done in FakeAVCeleb. However, considering the performance of the previous works, it is not a challenging dataset, and many experiments evaluated from this dataset gives very high AUC scores (>97). I suggest replacing it with some more challenging dataset to analyze to prevent statistical errors.
3. As the work is claiming the method preserve the universal knowledge in the ViT backbone, how is the data efficiency of this method? Does it work with small-scale training data? I guess the method might work considering the design, but I expect more analysis for this.
4. From the design, seems the architecture should work for other multimodal ViT based methods, is there any discovery for other domains for classification (multimodal emotion classification, etc)? Also, not just classifications, in deepfake area, some datasets expand the deepfake detections to spatial localization (ForgeryNet, VideoSham), and temporal localizations (AV-Deepfake1M, ForgeryNet). How’s the performance of this method working on these deepfake detection tasks?

Based on the idea above, I think the work contains reasonable motivation, method novelty and experiments. However, it is also important to see how the method performs for other scenarios which requires good universal knowledge of pretrained backbone. I think this work is in the borderline, and I’m happy to change to rating to accept based on the rebuttal.

**Suitability:**

3

---

### Official Review · Reviewer_7LBm · 2024-05-27

**Rating:** 4
**Confidence:** 3

**Summary:**

The article addresses the task of audio-visual forgery detection by incorporating prior knowledge through the use of a pre-trained Vision Transformer (ViT). Additionally, the AFI and ACI models are designed to align audio and video features. The experimental results demonstrate the effectiveness of the approach.

**Strengths:**

The article addresses the task of audio-visual forgery detection by incorporating prior knowledge through the use of a pre-trained Vision Transformer (ViT). Additionally, the AFI and ACI models are designed to align audio and video features. The experimental results demonstrate the effectiveness of the approach.

**Limitations:**

However, the innovation of the article seems somewhat lacking, as the proposed ACI module is essentially the same as the traditional cross-attention module, with the only difference being that the input signals are transformed into audio features and image features.

The method focuses on the consistency between audio and lip movements, which may limit the application scenarios of the algorithm.

**Suitability:**

2

---

### Official Review · Reviewer_Gtn4 · 2024-05-27

**Rating:** 4
**Confidence:** 4

**Summary:**

This paper proposed a forgery-aware Audio-distilled Multimodal Learning framework for deepfake detection, which utilizes forgery-aware injection which captures high-frequency discriminative features on both audio and visual signals and injects them into ViT. The experiments demonstrate the effectiveness of the proposed method.

**Strengths:**

1. The parameters of pretrained ViT are frozen to preserve its prior knowledge, which maximizes the performance of pre-trained VIT as much as possible.
2. The Audio-Distilled Cross-modal Interaction (ACI) bridges the domain gap of audio-visual modalities and can capture crucial audio-visual inconsistencies via cross-modal interaction.

**Limitations:**

1. It would be better to add some experiments to demonstrate the robustness of the method.

**Suitability:**

2

---

### Meta-Review · Area_Chair_mrH5 · 2024-07-02

**Recommendation:** Accept (Poster)
**Confidence:** 5

**Metareview:**

The paper presents a deepfakes detection method for audio-visual deepfakes based on ViT. The novelty aspect is using pre-trained ViT and injecting high frequency information into it through learning.

The reviewers have appreciated the method novelty. The rebuttal has mostly answered the queries.